# 3D AQI Mapping Data Assessment of Low-Altitude Drone Real-Time Air Pollution Monitoring

**Sarun Duangsuwan** [1,*] **, Phoowadon Prapruetdee** [1] **, Mallika Subongkod** [2] **and Katanyoo Klubsuwan** [3]

1   Electrical Engineering, Department of Engineering, King Mongkut's Institute of Technology Ladkrabang, Prince of Chumphon Campus, Chumphon 86160, Thailand; phoowadon.pr@kmitl.ac.th
2   Business Administration, Department of Business Administration, King Mongkut's Institute of Technology Ladkrabang, Prince of Chumphon Campus, Chumphon 86160, Thailand; mallika.su@kmitl.ac.th
3   Smart City Thailand Association (SCTA), Bangkok 10230, Thailand; katanyoo@eideas.co.th
*   Correspondence: sarun.du@kmitl.ac.th; Tel.: +66-99-1864411

**Abstract:** Air pollution primarily originates from substances that are directly emitted from natural or anthropogenic processes, such as carbon monoxide (CO) gas emitted in vehicle exhaust or sulfur dioxide ($SO_2$) released from factories. However, a major air pollution problem is particulate matter (PM), which is an adverse effect of wildfires and open burning. Application tools for air pollution monitoring in risk areas using real-time monitoring with drones have emerged. A new air quality index (AQI) for monitoring and display, such as three-dimensional (3D) mapping based on data assessment, is essential for timely environmental surveying. The objective of this paper is to present a 3D AQI mapping data assessment using a hybrid model based on a machine-learning method for drone real-time air pollution monitoring (Dr-TAPM). Dr-TAPM was designed by equipping drones with multi-environmental sensors for carbon monoxide (CO), ozone ($O_3$), nitrogen dioxide ($NO_2$), particulate matter ($PM_{2.5,10}$), and sulfur dioxide ($SO_2$), with data pre- and post-processing with the hybrid model. The hybrid model for data assessment was proposed using backpropagation neural network (BPNN) and convolutional neural network (CNN) algorithms. Experimentally, we considered a case study detecting smoke emissions from an open burning scenario. As a result, $PM_{2.5,10}$ and CO were detected as air pollutants from open burning. 3D AQI map locations were shown and the validation learning rates were apparent, as the accuracy of predicted AQI data assessment was 98%.

**Keywords:** 3D AQI mapping; BPNN–CNN model; smoke detection; open burning; Dr-TAPM; data assessment

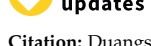



## 1. Introduction

Open burning is the burning of unwanted materials such as waste, tree branches, leaves, grass, and plastics; through open burning, smoke and emissions are released directly into the air. During open burning, air pollutants are the main considerations for health, human life, society, and cities. Globally, 5.4 million deaths were attributed to ambient air pollution in 2021. At 89.0%, low and middle-income countries had the highest portion of deaths. In a regional breakdown, it was found that the southeast Asian regions accounted for 943,000 deaths, while the highest number of deaths was 1,254,000 for the western Pacific region [1].

In Thailand, open burning is a major cause of air pollution, resulting in numerous public health problems and even death. It is a source of haze smoke in northern Thailand. Most open burning is caused by activities such as agricultural management, forest foraging, waste burning, and combustion. During the summer session from January to April, a high air quality index (AQI) above 300 is typically reported every year, meaning it is dangerous for human health. Smoke from the open burning of fields is a leading cause of the smog

crisis in many areas. The problem of smoke impacts socio-economic development, tourism, and public health in the daily lives of citizens in various regions [2].

The objective of air pollution monitoring is to fulfill statutory air quality reporting requirements [3]. An important step in developing an effective monitoring technology is to assess where air quality in the region is likely to deteriorate or if an area is at risk, such as in agricultural zones, industrial gas leakages, forest boundaries, open burning areas, and so on. These areas are impossible to monitor using human interaction through on-site surveys. In Thailand, one methodology to estimate air pollutant emissions is called the moderate-resolution imaging spectro-radiometer (MODIS) [4]. The MODIS solution is based on satellite image processing that can monitor a burned area with a 500 m × 500 m spatial resolution of the land surface. The monitoring results can show the spectrograph of air pollutants parameters such as $CO_2$, CO, $CH_4$, $NO_2$, $PM_{2.5}$, $PM_{10}$, and $NO_x$ in a two-dimensional (2D) mapping area. An alternative solution is to use unmanned aerial vehicles (UAV) or drones to monitor and survey the risk area or any location that has poor accessibility. Using a UAV mounted with multi-gas sensors provides the ability to fly to and survey an area to estimate the AQI by 3D mapping [5].

UAV-based air quality monitoring is challenging for research pioneers in terms of developing cost-effective UAV multi-functional monitoring for 3D AQI mapping. In the literature, the authors of [6] presented a self-driven drone and LoRaWAN network for air quality data assessment. The sensor boxset consisted of CO, $NO_2$, $SO_2$, $PM_{10}$, and $PM_{2.5}$ sensors, mounted on a rotary UAV. The advantage of the LoRaWAN network is that it enables long-range communication up to 5 km, through which all sensors send data back to the cloud network. Several works [7–12] have proposed the development of drone air pollution monitoring. In [7], the authors describe broad gas sensors for pollution such as $CO_2$, CO, $NH_3$, $SO_2$, $O_3$, and $NO_2$ to equip UAVs; the Zigbee module was used for sending data to the internet. Monitoring uses text messages and generates an air quality health index (AQHI) map. The AQHI map shows a specific region of measured data for each location in the text message. In the same work, the authors of [7,8] presented low-cost, off-the-shelf sensors and components that perform real-time monitoring of pollutants at low and high altitudes, through which the AQI can be monitored via mobile application for real-time data assessment. In [9], the authors presented various types of air quality sensors for CO, $SO_2$, $NO_2$, $O_3$, $PM_{2.5}$, $PM_{10}$, and black carbon (BC). They described the challenges regarding UAV-based systems, such as power consumption, weather conditions, equipment cost, and size. However, these related works did not consider 3D AQI mapping.

3D visualization and mapping are challenging for innovative air pollution monitoring. Existing work in [13] presents a spatial interpolation model for 3D AQI. The goal of spatial interpolation is to create a surface representing the 3D space as closely as possible. The results showed that the trilinear interpolation model is the most suitable for the 3D distribution of the air quality model in case studies. It can be seen that 3D virtualization and mapping employ post-data processing techniques based on the Gaussian plume model, which is a traditional method for air pollution monitoring. In [14], the authors presented the performance of the Gaussian plume model and neural network algorithm (GPM-NN) for the 3D AQI map. Further, the GPM-NN method was simulated by optimizing power consumption to increase the accuracy of AQI. Unfortunately, the limit of the GPM-NN method is that it is not designed to model dispersion under low wind conditions or sites close to the source, that is, at a distance less than 100 m. Additionally, trends of machine learning-based data prediction models are fulfilled for air quality monitoring.

Machine learning algorithms play an important role in new air pollution monitoring. Machine learning-based air pollution monitoring prediction models are statistical models that depend on observation and training data. Mostly, the main types of statistical models are regression models and neural network models. In [15], four types of machine learning algorithms—decision tree, random forest (RF), multilayer perceptron (MLP), and radial basis function (RBF) neural network—are presented to monitor the $PM_{2.5}$ data acquisition. The comparative results showed that the RBF neural network is the best suited algorithm

for predicting air quality models. In [16], nine machine learning algorithms—logistic regression (LR), linear discriminant analysis (LDA), K-nearest neighbors (K-NN), decision tree, Gaussian naïve Bayes (GNB), MLP, Adaboost (ADB), RF, and support vector machine (SVM)—are presented for classifying photogrammetry using a drone camera and LiDAR. The simulation result with the highest precision was the MLP algorithm. On the other hand, the result with the lowest precision was the GNB algorithm. Among the several machine learning methods, MLP-based neural networks have shown successful results, but there has been no consideration of these machine learning algorithms for 3D AQI mapping. 3D mapping to UAV was presented using a convolutional neural network (CNN) [17]. This method can utilize the UAV when lacking a global navigation satellite system (GNSS) signal. The CNN algorithm can transform the image from the UAV camera from RGB to a 3D construction and virtualization map. Then, the concept might be capable of designing 3D AQI mapping by UAV. In the case study of smoke detection [18,19], the CNN algorithm was used to classify the image of the smoke region situation for protection against forest fires. The 3D-CNN was developed to find the smoke target location [20]. Interestingly, the CNN algorithm was studied for air pollution prediction [21,22]. It is well-known that the CNN algorithm is suitable for image extraction and video prediction. In [23], the authors presented AQNet-based $PM_{2.5}$ monitoring. The proposed method uses a K-NN algorithm and deep neural network (DNN) procedure to jointly perform hybrid deep learning between the regression model and neural network model for 3D air quality.

As mentioned above, research on machine learning models has achieved significant progress. Although there are some related works on 3D virtualization maps, the proposed 3D AQI mapping has rarely been reported.

This paper presents a 3D AQI mapping data assessment for low-altitude drone real-time air pollution monitoring. The measurement of 3D AQI mapping data is assessed by using the BPNN–CNN algorithm (hybrid model). The objective of the proposed BPNN–CNN is to validate the dataset through training as a real-time monitor and to optimize the output layer of the results using 3D AQI mapping graphs on the *x*-axis, *y*-axis, and *z*-axis. Furthermore, the measurement of AQI using Dr-TAPM takes into account the real situation of open burning smoke detection in the present scenario.

## 2. Design of Frameworks

The framework for 3D AQI mapping is shown in Figure 1, where the procedures are divided into five sections. The AQI is measured remotely by the Dr-TAPM, while the cloud network is for data storage, data processing is for the pre-processing and post-processing of 3D AQI values using computer vision, the hybrid model is a proposed machine-learning model using the BPNN–CNN algorithm, and 3D AQI mapping is the output in 3D graphical form.

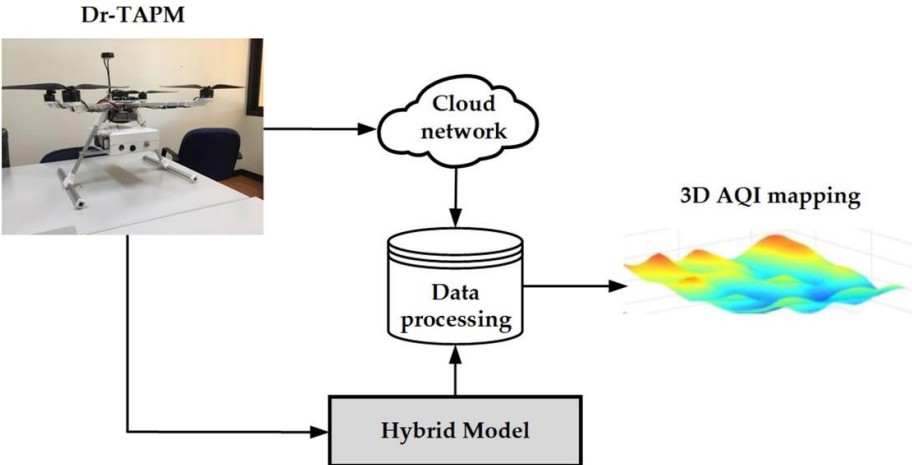

**Figure 1.** Framework for 3D AQI mapping data assessment.

*2.1. Dr-TAPM*

The Dr-TAPM platform is a rotary-wing UAV for air pollution monitoring. In previous work [6], the structure of the Dr-TAPM was designed using all aluminum to make it lightweight, low-cost, and robust. Figure 2 shows the structure and dimensions of the Dr-TAPM. The flight time is 30 min carrying a loadcell (sensor box), and the battery type is a lithium polymer (Li-Po) 6800 mAh 6 cell with two batteries. The weight is 2.3 kg without loadcell, and supports a loadcell up to 3 kg. The remote controller is WiFi 2.4 GHz with a range of 100 m, while the Pixhawk broad is the flight controller and flight positioning is by GPS module. Next, a sensor box can be equipped for flying, the sensor types of which are CO, $O_3$, $NO_2$, $PM_{2.5,10}$, and $SO_2$ sensors to measure the AQI value. An Arduino MEGA2260 controller board is used for control of sensor modules, and a Raspberry Pi 3 is used for data connection to the cloud network. Figure 3 shows a circuit diagram of all sensors and the controller boards in the sensor box. In this work, we confirm that all sensors were refreshed to their defaults before testing to ensure the return of accurate data.

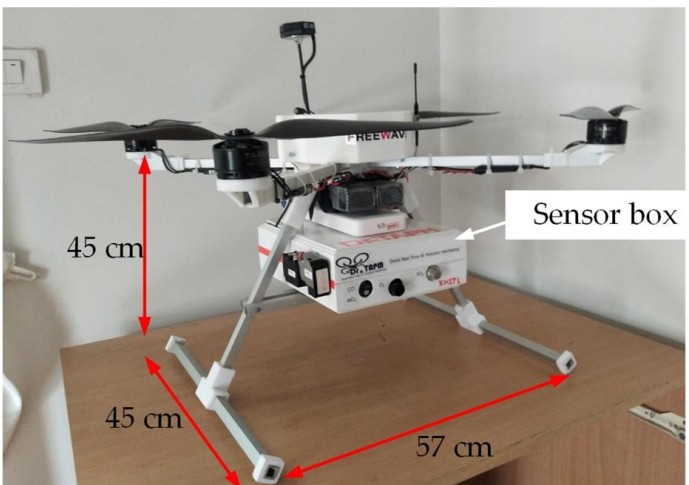

**Figure 2.** Dr-TAPM prototype.

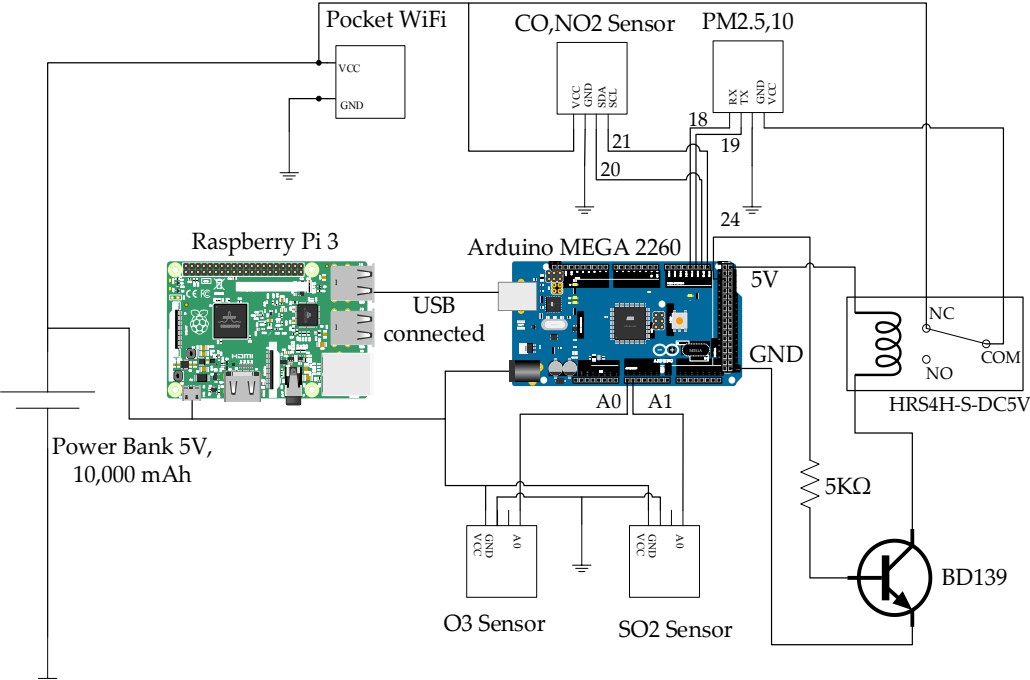

**Figure 3.** Circuit diagram of equipment inside the sensor box.

## 2.2. Cloud Network

The framework for the cloud network uses Google Firebase for real-time data storage and processing. To show the AQI monitoring, the mobile and web applications were displayed in real-time AQI status. When showing a green color from 0–50 levels, AQI is 'good', while the yellow color from 51–100 levels means 'moderate', the orange color from 101–150 levels means 'starting to impact health', the red color from 151–200 levels means 'unhealthy', and the purple color from 201–300 means 'very unhealthy'.

## 2.3. Data Processing

A framework for data processing was developed for comparing data between the measured AQI and the AQI predicted by the hybrid model. A computer-aided simulation is used to display the 3D AQI mapping in real-time from the Dr-TAPM surveying.

The design of the framework for data processing is based on high-level synthesis (HLS). HLS is an automated design process that interprets an algorithmic description of computer programming (MATLAB) and creates a model in a hardware description language (HDL). Our framework presents the development steps, which are illustrated in Figure 4. Steps that involve software, middleware, and hardware have been depicted with different shades of color. The first step of the framework is to take real-time AQI raw data and convert it into HLS language. To generate the HLS code describing the software-based MATLAB program, the Simulink tool provides a set of HLS functions that diagnose the raw data and the data predicted by the hybrid model. In the next step, the HLS code is subsequently translated to HDL code by the Simulink tool, which takes into account optimization directives such as plotting the 3D AQI mapping and then displays the results in mobile and web applications.

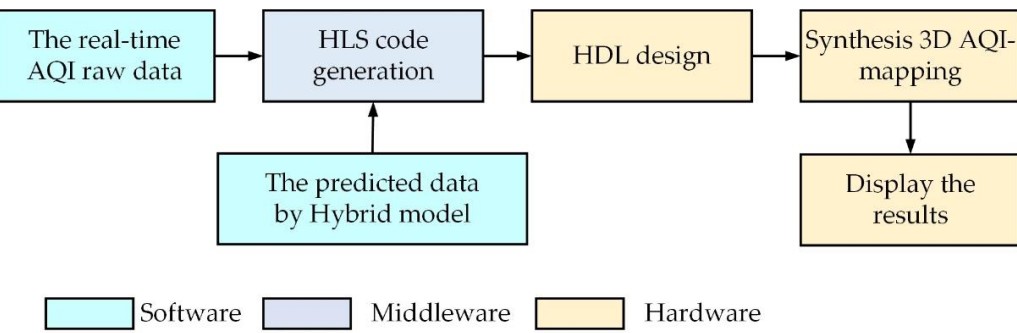

**Figure 4.** Data processing of synthesis 3D AQI mapping.

## 3. Methodology

*Hybrid Model*

Accurate air pollution monitoring by UAV has difficulty precisely predicting 3D AQI for a region at a specific time. The machine learning model fills the gap between the capability of the Dr-TAPM and 3D AQI mapping. Computer vision features in CNNs [17,22] have been used to construct 3D map imagery. A BPNN [24–26] is a prediction method for accurate air pollutant values in real-time monitoring. The motivation for this work is to present a hybridization approach (BPNN–CNN) using BPNN and CNN to produce reliable and accurate 3D AQI mapping with the low-altitude Dr-TAPM. Figure 5 shows a schematic diagram of the hybrid model using BPNN and CNN algorithms, and the graphical illustration of the proposed hybrid framework is shown in Figure 6. In Figure 6, the BPNN is a fully connected neural network with three layers: one input layer, one hidden layer, and one output layer. The steps of the BPNN consist of two main propagations. Firstly, there is the feedforward propagation and weight adjustment. In the feedforward propagation, AQI raw data are input signals that are assigned a weight by the rectified linear unit (ReLU) function or activation function in the hidden layer; then, the weight is sent to the output layer to calculate the predicted AQI values. Secondly, the backpropagation is a procedure of optimization between the predicted data and tested data by finding

the error indicators. After the data is predicted, the next step is a CNN procedure with three layers: the feature extraction layer, the convolution layer, and the 3D feature map in different colors.

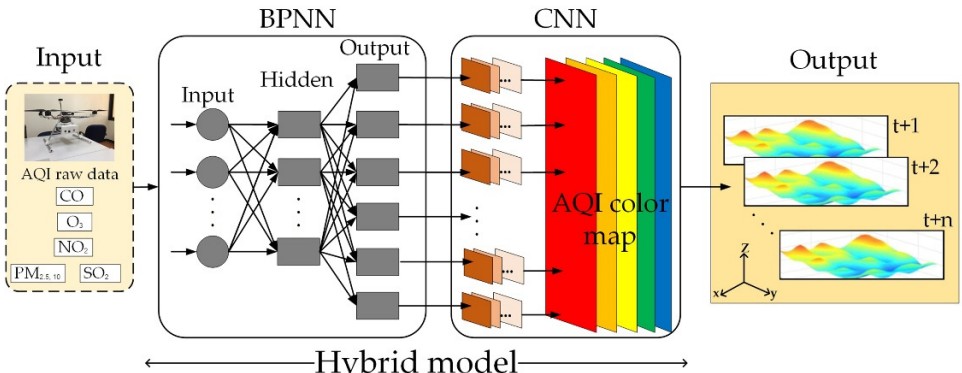

**Figure 5.** A schematic diagram of the hybrid model.

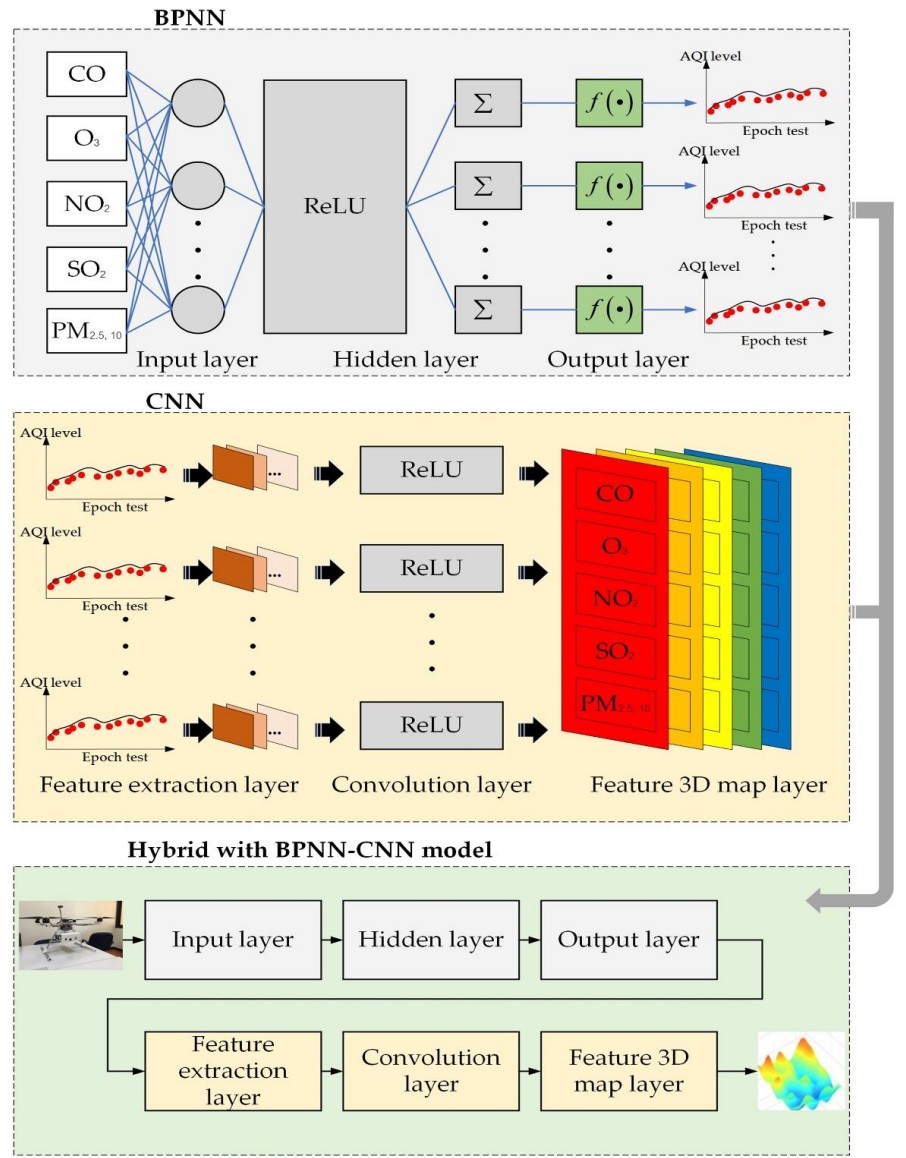

**Figure 6.** A schematic diagram of the hybrid model using BPNN–CNN.

The formulation of the BPNN algorithm was proposed by R. Hecht-Nielsen in 1992. It can be seen as an extension of the perceptron algorithm [27]. A general formula for a single layer is as follows:

$$y = f\left(\mathbf{w}^T\mathbf{x} + b\right) \tag{1}$$

where $\mathbf{x}$ is the test data in the input layer, $y$ is the output layer, $\mathbf{w}$ is the weight layer, and $b$ is the bias. The activation function is a control weight and bias to estimate the backpropagation error as a minimum by using the sigmoid function or ReLU function. The most widely used activation functions are the following:

$$f(x) = \frac{1}{1 + e^{-l}} \tag{2}$$

$$\text{ReLU} = \begin{cases} l & l > 1 \\ 0 & l \leq 0 \end{cases} \tag{3}$$

where $l$ is sub-dataset.

The hidden layer in Figure 6 denotes a weighted connection by using the ReLU unit to activate the data input. The training model of the BPNN is optimized by using stochastic gradient descent (SGD), which is an optimization method for reducing the loss function or backpropagation error for weight and control bias simultaneously. For the training model, the output unit activation function is

$$\hat{y} = f\left(\mathbf{w}^T\mathbf{h} + b\right) \tag{4}$$

where $\mathbf{h}$ is the training model using SGD.

The accuracy of the backpropagation error indicators or learning rates is shown by the mean absolute error (MAE), the root mean square error (RMSE), and the coefficient of determination ($R^2$). The formula for the three indicators is given by

$$\text{MAE} = \sum_{i=1}^{K} \frac{|y_i - \hat{y}_i|}{K} \tag{5}$$

$$\text{RMSE} = \sqrt{\sum_{i=1}^{K} \frac{(y_i - \hat{y}_i)^2}{K}} \tag{6}$$

$$R^2 = \frac{\sum\limits_{i=1}^{K} y_i\hat{y}_i - \sum\limits_{i=1}^{K} y_i \sum\limits_{i=1}^{K} \hat{y}_i}{\sqrt{\sum\limits_{i=1}^{K} y_i^2 - \left(\sum\limits_{i=1}^{K} y_i\right)^2} \cdot \sqrt{\sum\limits_{i=1}^{K} \hat{y}_i^2 - \left(\sum\limits_{i=1}^{K} \hat{y}_i\right)^2}} \tag{7}$$

where $K$ denotes the number of total raw datasets.

The formulation of the CNN algorithm was proposed by LeCun et al. in 1989. The function of a CNN is similar to a feedforward neural network in that they both use a convolution operation to extract the features of input data. The main functions of a CNN are divided into the feature extraction layer, the convolution layer, and the feature map. In feature extraction, the input data is filtered before feeding the data sequence into the convolution layer. Then, the convolution layer is activated by using the ReLU unit. In general, this is a max-pooling layer. The max-pooling consists of two steps. Firstly, it makes the 2D-convolution of the predicted data from the BPNN, which is useful when determining the 3D map in the next layer. Secondly, the pooling layer helps condense the size of the output layer. The last layer is the 3D feature map. This process translates the AQI values for each color level and map by the context of the display as follows

$$S(\hat{y}, K) = \sum_n \sum_m I(x, y, z) C(\hat{y} - n, K - m) \tag{8}$$

where $I(x, y, z)$ denotes the 3D AQI map levels from 1 to 300 index in each color—green, yellow, orange, red, and purple—where $n$ and $m$ are independent of the pollutant parameters and the locations being tested. $C$ is the function of the convolution layer from the max-pooling step, which depends on the predicted data.

## 4. Data Assessment

### 4.1. Training Model

The dataset is separated into two parts. Firstly, 20% of the actual data were used for testing. These data are $\mathbf{x} = [x_1, x_2, x_3, \dots, x_n]$, where $n$ represents the pollutant sensors such as the CO, $O_3$, $NO_2$, $PM_{2.5,10}$, and $SO_2$ sensors. Secondly, 80% of the data were used for training the BPNN–CNN model (See Table 1).

**Table 1.** Parameters for training the BPNN–CNN model.

| Parameters | Values |
| --- | --- |
| Number of $K$ raw datasets raw datasets | 1400 |
| Number of $n$ | 5 |
| Number of $m$ | 40 |
| Testing data | 20% |
| Training data | 80% |

The framework of the training model was established using Python programming in Scikit-learn and TensorFlow. Scikit-learn is a feature operation of a machine-learning model that can provide a hyperparameter setting, facilitating the construction of a BPNN model according to library requirements. TensorFlow is a tool for the construction of a CNN to offer image processing in 3D graphs. To validate the training model, the four parameter sets for the BPNN–CNN model include input layers, hidden layers, the activation function, and an optimizer. The input layers were set at seven nodes and one hidden layer, with each layer set as 16 nodes. The activation function was a ReLU unit, and an SGD was provided as the optimizer.

### 4.2. Experimental Setup in Case Study of Open Burning Smoke Detection

In this work, we consider a case study of open burning, which allows the Dr-TAMP to detect smoke-based air pollutants using remote sensing. Figure 7a shows a schematic model of the experimental setup on the *x*-axis, *y*-axis, and *z*-axis. The UAV height was set to 5 m and 10 m, and the steps up were from 1 m to 20 m. For the open burning point source, we used the burning of agricultural waste materials such as palm wood, unwanted trees, leaves, grass, and dry weeds. Figure 7b shows the field test at King Mongkut's Institute of Technology Ladkrabang Prince of Chumphon Campus. We note that the surrounding environment is clear and rarely impacted by the wind. The data was captured 30 times per position and recorded into the cloud network every 10 s. Our experimental data were recorded as a total of 1400 datasets for training the BPNN–CNN model.

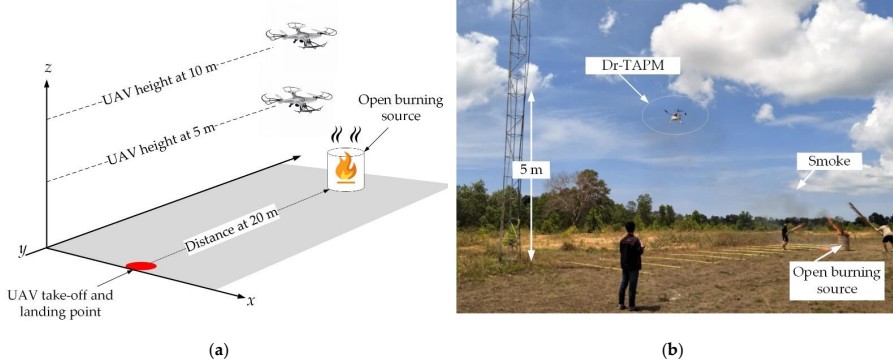

(a)                                              (b)

**Figure 7.** (**a**) Schematic model setup in 3D; (**b**) open burning smoke detection in the field test.

### 4.3. Results

This subsection will report the experimental results when considering two altitudes level of the Dr-TAPM. The result of all air pollutant parameters at 5 m altitude are shown in Figure 8a–j, while the parameters at 10 m altitude are shown in Figure 9a–j. The 3D AQI maps are depicted in Figure 10a,b.

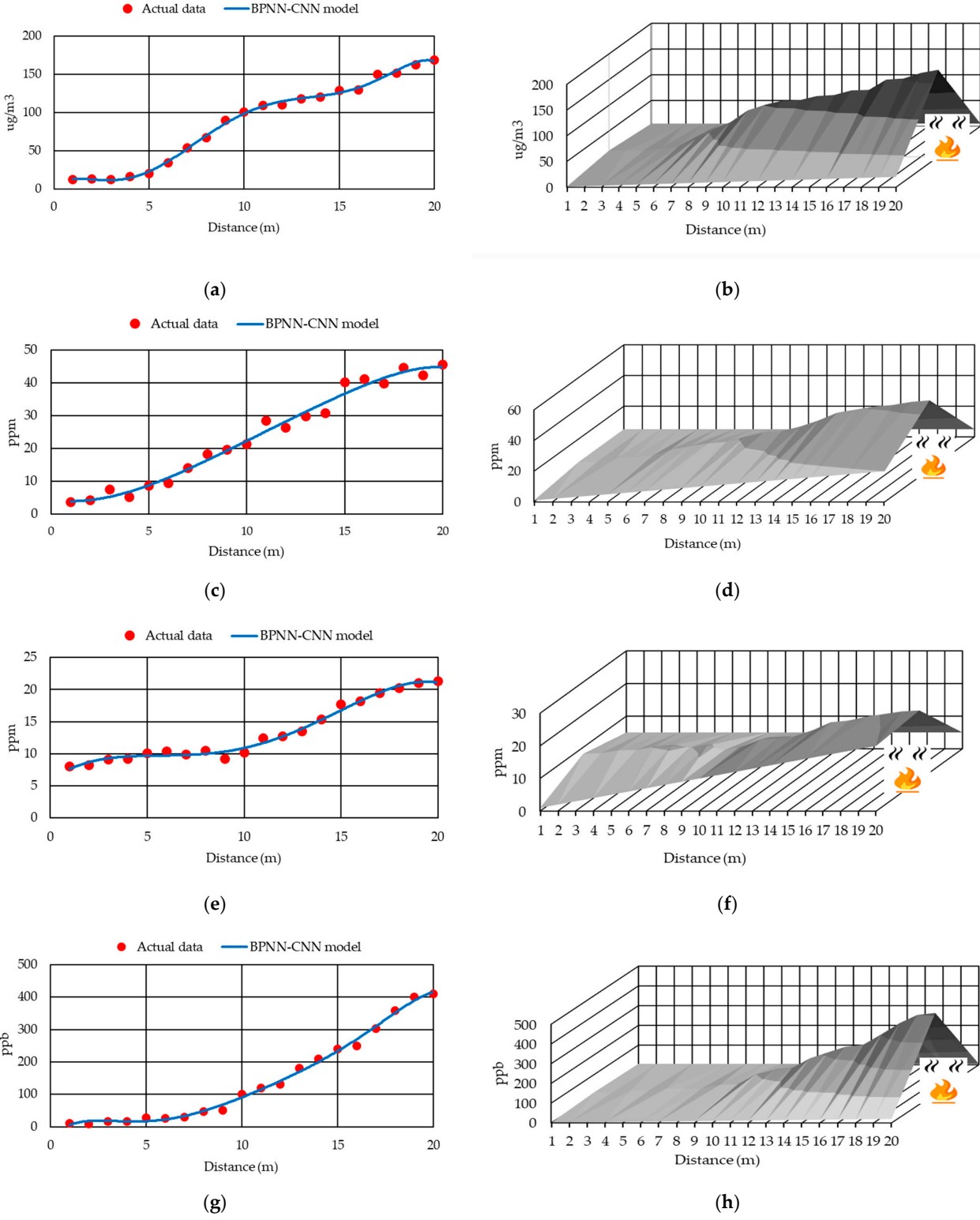

**Figure 8.** *Cont.*

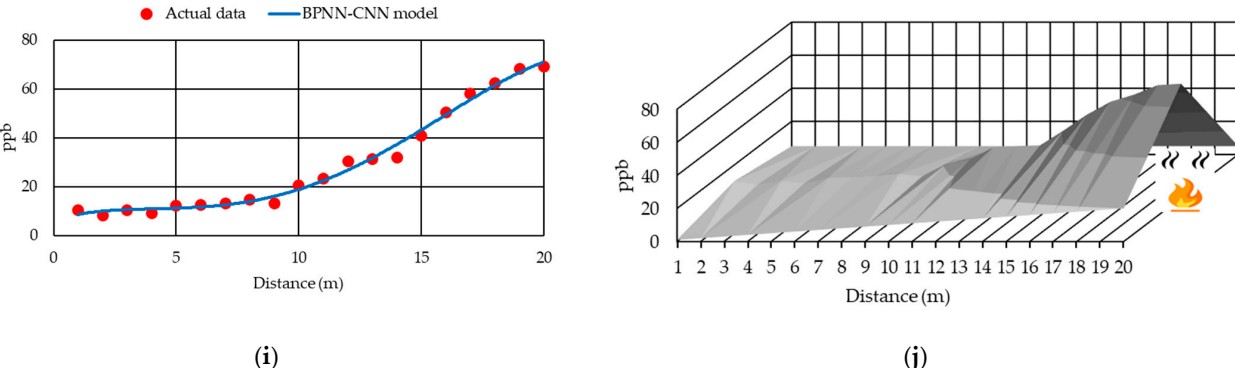

(**i**)　　　　　　　　　　　　　　　　　　(**j**)

**Figure 8.** Air pollutant parameters at 5 m altitude: (**a**) $PM_{2.5,10}$ in 2D; (**b**) $PM_{2.5,10}$ in 3D; (**c**) CO in 2D; (**d**) CO in 3D; (**e**) $O_3$ in 2D; (**f**) $O_3$ in 3D; (**g**) $SO_2$ in 2D; (**h**) $SO_2$ in 3D; (**i**) $NO_2$ in 2D; (**j**) $NO_2$ in 3D.

(**a**)　　　　　　　　　　　　　　　　　　(**b**)

(**c**)　　　　　　　　　　　　　　　　　　(**d**)

(**e**)　　　　　　　　　　　　　　　　　　(**f**)

**Figure 9.** *Cont.*

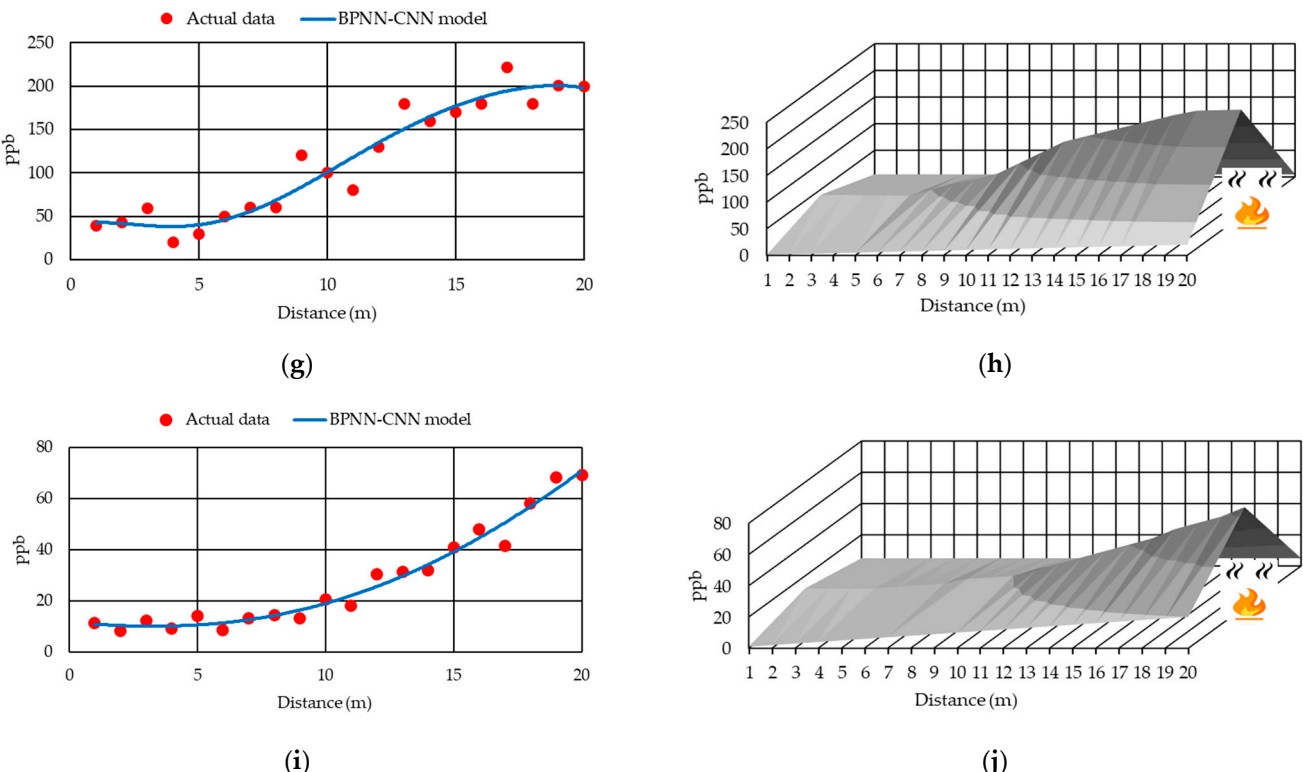

**Figure 9.** Air pollutant parameters at 10 m altitude: (**a**) PM$_{2.5,10}$ in 2D; (**b**) PM$_{2.5,10}$ in 3D; (**c**) CO in 2D; (**d**) CO in 3D; (**e**) O$_3$ in 2D; (**f**) O$_3$ in 3D; (**g**) SO$_2$ in 2D; (**h**) SO$_2$ in 3D; (**i**) NO$_2$ in 2D; (**j**) NO$_2$ in 3D.

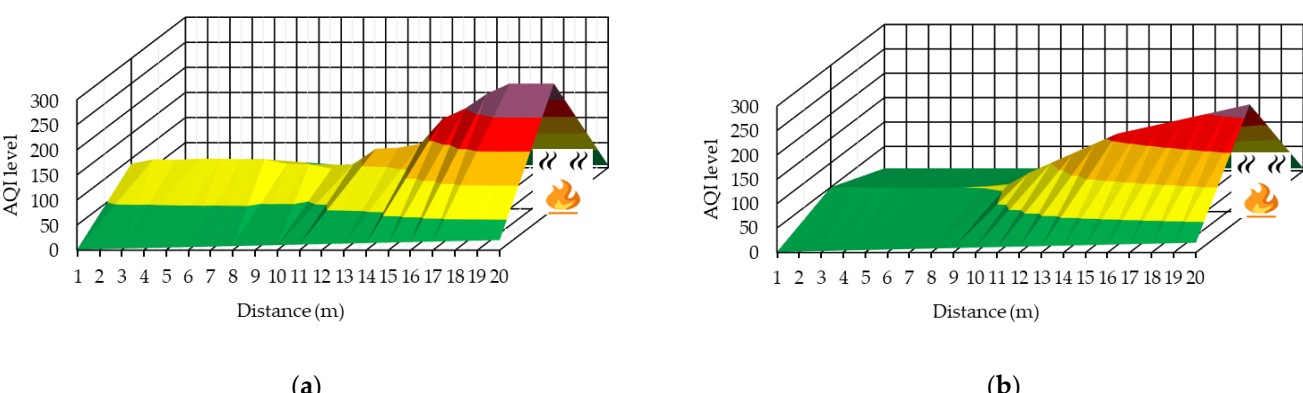

**Figure 10.** 3D AQI map location in color (**a**) at 5 m of UAV altitude; (**b**) at 10 m of UAV altitude.

Figure 8a shows the result at 5 m of UAV altitude for the measurement data plot of PM$_{2.5,10}$, with the actual data and the curve predicted by the BPNN–CNN model. The actual data of the PM$_{2.5,10}$ level varied from 20 ug/m$^3$ to 160 ug/m$^3$, while the BPNN–CNN model can use the real-time monitoring of actual data to create a smoothed curve. Then, the predicted data model can perform modeling of the 3D mapping result for PM$_{2.5,10}$, as shown in Figure 8b. We can see that the PM$_{2.5,10}$ curve increased slightly from the distance of 10 m to 20 m. Figure 8c,d shows the actual data plot of CO emissions from the open burning source. The level of CO was 5.5 ppm to 45 ppm, according to the BPNN–CNN learning curve. It is well known that the CO must be lower than 50 ppm for the safety of health. The ozone (O$_3$) level is shown in Figure 8e,f. The BPNN–CNN curve model predicted the actual O$_3$ data from 6.3 ppm to 22 ppm. It can be seen that the ozone also increased from 10 m distance to 20 m. Generally, ozone must not be above 10 ppm, below which it is not a risk for health. Next, sulfur dioxide gas (SO$_2$) is shown in Figure 8g,h. The

result from the BPNN–CNN model curve varies from 40 ppb to 400 ppb. In practice, $SO_2$ is dangerous for health if the level is higher than 100 ppb [3]. It can be seen that the result was higher than 100 ppb from 10 m to 20 m. Additionally, nitrogen dioxide gas ($NO_2$) is shown in Figure 8i,j, with the results ranging from 12 ppb to 65 ppb. It should be noted that the $NO_2$ must be not above 20 ppb [3]. When the UAV steps were close to the open burning source, then the result was higher than 20 ppb. We can see that both $SO_2$ and $NO_2$ gases are dangerous when sensed close to the open burning source.

The air pollutant results at 10 m of UAV altitude are shown in Figure 9a–j. The results are similar to those at 5 m. To distinguish the measurement data, it can be seen that the actual data is distributed and scattered more than previously. However, the data predicted by the BPNN–CNN model learns the uncertain data in real-time. Figure 9a,b shows $PM_{2.5,10}$, where the BPNN–CNN curve was detected from 18 ug/m$^3$ to 90 ug/m$^3$. It is well known that $PM_{2.5,10}$ should not be above 50 ug/m$^3$. We observe that the smoke detection by Dr-TAPM captured $PM_{2.5,10}$ over 50 ug/m$^3$ at 12 m to 20 m near an open burning source. CO is shown in Figure 9c,d where the concentration was from 5 ppm to 36 ppm. Ozone ($O_3$) was detected at 7–22 ppm, as shown in Figure 9e,f. $SO_2$ gas was detected at 48 ppb to 200 ppb, and $NO_2$ gas at 10 ppb to 70 ppb, as shown in Figure 9g–j.

It is apparent that the smoke from open burning resulted in highly intensive pollution to the air. In particular, $PM_{2.5,10}$ and CO were the main parameters of air pollution in this experiment. The 3D AQI map location for both levels of the UAV monitor is shown in Figure 10a,b. The separation of color on the 3D AQI map shows the air pollution in the case study for the open burning scenario.

The accuracy of the BPNN–CNN model is shown with MAE, RMSE, and $R^2$, in Tables 2 and 3. We observe that the average accuracy of the proposed BPNN–CNN model was 0.992 in the 5 m UAV altitude test, and 0.965 at 10 m, measured by $R^2$. Therefore, the average accuracy by $R^2$ in the two UAV altitude scenarios was 0.979 or 98% as a percentage of the precision learning rate.

**Table 2.** Learning rate between the actual data and BPNN–CNN model in the 5 m altitude test.

| Air Pollutant Parameters | MAE | RMSE | $R^2$ |
|:---:|:---:|:---:|:---:|
| $PM_{2.5,10}$ | 0.351 | 1.561 | 0.995 |
| CO | 0.510 | 2.054 | 0.983 |
| $O_3$ | 0.352 | 1.565 | 0.994 |
| $SO_2$ | 0.310 | 1.341 | 0.997 |
| $NO_2$ | 0.250 | 1.118 | 0.991 |
| Average | 0.402 | 1.787 | 0.992 |

**Table 3.** Learning rate between the actual data and the BPNN–CNN model in the 10 m altitude test.

| Air Pollutant Parameters | MAE | RMSE | $R^2$ |
|:---:|:---:|:---:|:---:|
| $PM_{2.5,10}$ | 1.050 | 4.696 | 0.964 |
| CO | 0.950 | 4.248 | 0.968 |
| $O_3$ | 1.100 | 4.919 | 0.961 |
| $SO_2$ | 1.150 | 5.143 | 0.956 |
| $NO_2$ | 0.850 | 3.801 | 0.979 |
| Average | 1.020 | 4.561 | 0.965 |

## 5. Conclusions

This paper presented a 3D AQI mapping data assessment for low-altitude drone real-time air pollution monitoring in a case study of smoke detection from an open burning scenario. The data assessment was performed successfully using Dr-TAPM as the mobility tool for operational air pollutant monitoring. The air pollutant parameters that were considered were carbon monoxide (CO), ozone ($O_3$), nitrogen dioxide ($NO_2$), particulate matter ($PM_{2.5,10}$), and sulfur dioxide ($SO_2$). Furthermore, the proposed BPNN–CNN model

predicted the results of air pollution parameters and generated 3D AQI mapping locations. It was shown that the proposed BPNN–CNN model could assess air pollution monitoring at 98% accuracy for data assessment in an open burning scenario. The results of this work benefit the design and development of a low-altitude UAV for ground truth air pollution monitoring based on 3D remotely sensed AQI data-driven processing. In future work, path planning and 3D AQI trajectory mapping will be presented and the consideration of large-scale scenarios, flight operation time, and multiple open burning sources will be investigated.

**Author Contributions:** Conceptualization, S.D. and K.K.; methodology, S.D.; software, S.D.; validation, S.D., P.P., M.S. and K.K.; investigation, S.D.; writing—original draft preparation, S.D.; writing—review and editing, S.D., P.P., M.S. and K.K. All authors have read and agreed to the published version of the manuscript.

**Funding:** The research on 3D AQI Mapping Data Assessment of Low-Altitude Drone Real-Time Air Pollution Monitoring by King Mongkut's Institute of Technology Ladkrabang/ Prince of Chumphon Campus received funding support from the National Science, Research and Innovation Fund (NSRF) under Grant No. RE-KRIS/FF65/50.

**Data Availability Statement:** The data presented in this study are available on request to the corresponding author.

**Conflicts of Interest:** The authors declare no conflict of interest.

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
