# Peer review of "3D AQI Mapping Data Assessment of Low-Altitude Drone Real-Time Air Pollution Monitoring"

_drones, doi:10.3390/drones6080191_

Round 1

Reviewer 1 Report

This study is focus on drone application on air quality monitoring, the 3D AQI mapping might be a powerful tool for environmental management. The manuscript provide sufficient data and analysis. I think this manuscript is suitable for publication after some improvements.

Here are some suggestions for the authors’ reference:

1. The concentration of air pollutant could be influenced greatly by meteorological condition, such as wind and rainfall. So how can the Dr-TAPM platform maintain accurate if the burning is disturbed by complicated weather?

2. Line 34-40: Where or what references is the number of deaths attributed to ambient air pollution come from?

3. Line 41-49: Add references about the open burning in Thailand.

4. Line 308-311: Please provide the reference or standard why SO2 and NO2 should below 100 and 20 ppb. Same as PM2.5 and PM10 in Line 321.

5. The authors can show the prospect in the field of environmental management in detail, for AQI is general used to evaluate public health risks from air pollution in a relative large area.

Author Response

Dear reviewer 1, 

We have attached the reply report in word, all comments have been carefully rechecked one by one. 

Thank you very much for your kind comments.

Reviewer 2 Report

The article having titled as “3D AQI Mapping Data Assessment using BPNN-CNN Model 2 of Low-Altitude Drone Real-Time Air Pollution Monitoring”, the above article is lie in my area of expertise that’s why I recommend the following changes in the manuscript.

1.       The title is very long as well as if other than the expert of this area wants to read, they neglect to read due to complex words used in title. My suggestions are to rewrite the title of article short and with out the algorithm names. However, authors can use the algorithm names in abstract.

2.       What is the motivation of this research and what is the major outcomes?

3.       Must write the improvement in terms of percentage by comparing with other methods.

4.       In introduction its better to write some sort of case study

5.        What is the main problem statement and proposed solution of this article? Must add a heading other than methodology

6.       Size of the images mainly figure 5 are very small its best to redraw and put

7.       Its best if author make another heading of system architecture and put figure 6 in it and elaborate it

8.       I am not satisfied with the given results its best to upload the video at youtube and any other platform and put the link inside the paper.

9.       In conclusion where is the data set of this article?

Author Response

Dear reviewer 2, 

We have attached the reply report in word, all comments have been carefully rechecked one by one. 

Thank you very much for your kind comments.

Reviewer 3 Report

The present paper addresses a very important topic: mapping of air pollution carried out here taking advantage of a double approach, i.e. implementing an air pollution sensing unit combined with a machine learning method.

The idea is interesting , the paper well written and the approach clearly comunicated. 

There are two major issues in the paper which are related to the experimental part.

Indeed, in the paper a spot polluting source is considered, and the UAV is operated in the close proximity of such source. Such condition is very helpful and easy to map, but on the other hand is not reflecting the tyipcal conditions of a polluted area, where :

the UAV is normally not necessarily close to the source (which is often unknown) and thus pollutant concentrations are lower

- many phenomena (wind, diffusion,...) complicate the concentration gradients, producing unhomogeneous pollutant distributions in the area

- the pollutants from one source are being mixed with pollutant from other sources in the neighbourhood 

- in the case of large areas the autonomy of the drone is a problem and the battery change is needed: flight discontinuity complicate very much pollutant mamping. 

As a consequence, to be validated, the method and the machine learning algorithms should be verified in more complicated scenarios, characterized by more complicated and unhomogeneous conditions. 

The second problem is related to the validation: the authors eventually producee some nice pollutants maps, but are such distribution "true", i.e. corresponding to the actual distribution? I believe a ground truth is missing , to validate the method. 

OTHER COMMENTS:

It is not clear how much air flow caused by the UAV are affecting the pollutant distribution and how the machine learning algorithm is compensating such "noise" introduced by the UAV itself. Please discuss in the paper. 

How is the method performing in the case of lower concentrations or at higher distances from the source?

Provide details about flight operation: speed, autonomy (was sufficient one flight to map the area?)

Improve the quality of figures (in particular use larger fonts in figures 5 and 7 but also 4 and 6)

Author Response

Dear reviewer 3, 

We have attached the reply report in word, all comments have been carefully rechecked one by one. 

Thank you very much for your kind comments.

Reviewer 4 Report

General comments

The paper entitled “3D AQI Mapping Data Assessment using BPNN-CNN Model of Low-Altitude Drone Real-Time Air Pollution Monitoring” treats about a topic of the highest interest in the air automatic pollution detection scope. Authors used a hybrid method to accomplish the declared task, namely BPNN-CNN. Those methods are not new; therefore the novelty of the manuscript concerns the application of these two neural network methodologies to the air pollution scope.

The used language is obscure and full of typos and inaccuracies, in brief hard to follow. The manuscript needs a deep revision from an English native speaker to improve its quality until to reach a full readability. The paper is linear and well-structured. The typographical outline is nearly sufficient. Key Words are pertinent to the paper content and appropriate. The highlights are not included within the paper, therefore no judgement has been provided about them. The References section is rich both in quantity and quality. Graphic representations are all of good quality and self-explaining. Finally, the treated topic falls under the journal scope without any doubt. This reviewer considers the paper worthy of publication after major changes.

Specific comment

Line 14: please, change “particular matter” into “particulate matter”.

Line 19: Authors introduce the acronym AQi without explaining it.

Line 38: Please, change “midden” into “middle”.

Line 48: please change “impacts” into “impacts on”.

Line 92-93: the sentence is unclear should be improved.

Line 97: please, change “had become” into “played”.

Line 98-99: the sentence is not accurately written, please improve it.

Line 111-112: the sentence appears to be obscure, please improve it.

Line 116: please, change “might capable” into “might be capable”.

Line 172: please, change “for comparison” into “for comparing”

Etc etc…..

Author Response

Dear reviewer 4, 

We have attached the reply report in word, all comments have been carefully rechecked one by one. 

Thank you very much for your kind comments.

Round 2

Reviewer 2 Report

Its an average paper. Moreover, authors did not submit the compliance reply.

Author Response

Thank you for your comments, we have rechecked by following the comments in the 2nd major revision. 

Reviewer 3 Report

The authors have only prtially answered some of the comments from my previous reviews. Also for some of the comments thee authors have just answered the referee, without correcting the paper. 

Please improve the paper according to the different comments, including the major ones. 

Please highlight in yellow the corrected parts. 

Please find here below again my previous review. 

There are two major issues in the paper which are related to the experimental part.

Indeed, in the paper a spot polluting source is considered, and the UAV is operated in the close proximity of such source. Such condition is very helpful and easy to map, but on the other hand is not reflecting the tyipcal conditions of a polluted area, where :

the UAV is normally not necessarily close to the source (which is often unknown) and thus pollutant concentrations are lower

- many phenomena (wind, diffusion,...) complicate the concentration gradients, producing unhomogeneous pollutant distributions in the area

- the pollutants from one source are being mixed with pollutant from other sources in the neighbourhood 

- in the case of large areas the autonomy of the drone is a problem and the battery change is needed: flight discontinuity complicate very much pollutant mamping. 

As a consequence, to be validated, the method and the machine learning algorithms should be verified in more complicated scenarios, characterized by more complicated and unhomogeneous conditions. 

The second problem is related to the validation: the authors eventually producee some nice pollutants maps, but are such distribution "true", i.e. corresponding to the actual distribution? I believe a ground truth is missing , to validate the method. 

OTHER COMMENTS:

It is not clear how much air flow caused by the UAV are affecting the pollutant distribution and how the machine learning algorithm is compensating such "noise" introduced by the UAV itself. Please discuss in the paper. 

How is the method performing in the case of lower concentrations or at higher distances from the source?

Provide details about flight operation: speed, autonomy (was sufficient one flight to map the area?)

Improve the quality of figures (in particular use larger fonts in figures 5 and 7 but also 4 and 6)There are two major issues in the paper which are related to the experimental part.

Indeed, in the paper a spot polluting source is considered, and the UAV is operated in the close proximity of such source. Such condition is very helpful and easy to map, but on the other hand is not reflecting the tyipcal conditions of a polluted area, where :

the UAV is normally not necessarily close to the source (which is often unknown) and thus pollutant concentrations are lower

- many phenomena (wind, diffusion,...) complicate the concentration gradients, producing unhomogeneous pollutant distributions in the area

- the pollutants from one source are being mixed with pollutant from other sources in the neighbourhood 

- in the case of large areas the autonomy of the drone is a problem and the battery change is needed: flight discontinuity complicate very much pollutant mamping. 

As a consequence, to be validated, the method and the machine learning algorithms should be verified in more complicated scenarios, characterized by more complicated and unhomogeneous conditions. 

The second problem is related to the validation: the authors eventually producee some nice pollutants maps, but are such distribution "true", i.e. corresponding to the actual distribution? I believe a ground truth is missing , to validate the method. 

OTHER COMMENTS:

It is not clear how much air flow caused by the UAV are affecting the pollutant distribution and how the machine learning algorithm is compensating such "noise" introduced by the UAV itself. Please discuss in the paper. 

How is the method performing in the case of lower concentrations or at higher distances from the source?

Provide details about flight operation: speed, autonomy (was sufficient one flight to map the area?)

Improve the quality of figures (in particular use larger fonts in figures 5 and 7 but also 4 and 6)

Author Response

Thank you very much for your comments all very useful. We have carefully rechecked following the comments. Thank again for fulfilling the manuscript.   

Reviewer 4 Report

After the reading of this second version of the manuscript, this reviewer recognizes the efforts brought by Authors to improve in deep their work. Therefore, the manuscript can be published in the present form.

Author Response

Thank you for your comments and fulfill the manuscript. We hope to you that this research will be opened a good opportunity for us. Thank you. If you have any suggestions, please contact the corresponding author email at  [email protected]      

Round 3

Reviewer 3 Report

The paper has been properely improved and is now ready for publication